

# Understanding the black-box: towards interpretable and reliable deep learning models

Tehreem Qamar and Narmeen Zakaria Bawany

Center for Computing Research, Department of Computer Science and Software Engineering, Jinnah University for Women, Karachi, Pakistan

## ABSTRACT

Deep learning (DL) has revolutionized the field of artificial intelligence by providing sophisticated models across a diverse range of applications, from image and speech recognition to natural language processing and autonomous driving. However, deep learning models are typically black-box models where the reason for predictions is unknown. Consequently, the reliability of the model becomes questionable in many circumstances. Explainable AI (XAI) plays an important role in improving the transparency and interpretability of the model thereby making it more reliable for real-time deployment. To investigate the reliability and truthfulness of DL models, this research develops image classification models using transfer learning mechanism and validates the results using XAI technique. Thus, the contribution of this research is twofold, we employ three pre-trained models VGG16, MobileNetV2 and ResNet50 using multiple transfer learning techniques for a fruit classification task consisting of 131 classes. Next, we inspect the reliability of models, based on these pre-trained networks, by utilizing Local Interpretable Model-Agnostic Explanations, the LIME, a popular XAI technique that generates explanations for the predictions. Experimental results reveal that transfer learning provides optimized results of around 98% accuracy. The classification of the models is validated on different instances using LIME and it was observed that each model predictions are interpretable and understandable as they are based on pertinent image features that are relevant to particular classes. We believe that this research gives an insight for determining how an interpretation can be drawn from a complex AI model such that its accountability and trustworthiness can be increased.

# INTRODUCTION

Artificial intelligence (AI) in the form of deep learning (DL) models has gained significant advancement in recent years (*Yi et al., 2016*). We are increasingly dependent on artificial intelligence, more specifically on the deep learning for almost everything we do. From recommendations for shopping to self-driving cars, from loan approval to face detection, our lives are affected by these AI based systems. Deep learning involves deep neural networks (DNNs) that learn in layers and resemble the human brain. They become the central technology and have been extensively utilized across various domains such as finance, medicine, natural language processing, cyber security, bioinformatics, robotics, *etc.*

Corresponding author
Tehreem Qamar,
tq.tehreem@gmail.com

Unlike traditional machine learning models, DL models have the capability to automatically engineer features; therefore, there is no need for explicit feature extraction with human supervision. DL enables learning and classification in a single shot as they implicitly examine data and look for features that correlate and save days or even months of work of data scientists' and researchers' by identifying new, more complicated features that they would overlook (*Lecun, Bengio & Hinton, 2015*). Additionally, DL models are able to be trained on unstructured, unlabeled data and produce ample accuracy. On the other hand, DL models at the core are black-boxes *i.e.,* their decisions are hidden in the thousands of simulated neurons, grouped into dozens or hundreds of highly interconnected layers. Further, the back-propagation method updates the computations made by individual neurons so that the network may minimize the loss function thereby improving the performance of the model. Consequently, the DL models become more complicated resulting in the high performing black-box models (*Minaee et al., 2021*; *Hohman et al., 2019*). Deep learning models are undergoing continuous development and recently surpassed human performance on tasks such as image classification (*Alzubaidi et al., 2021*).

Image classification is one of the challenging and critical tasks in today's AI systems, implemented by utilizing convolutional neural network (CNN), the most popular DL model. CNN identifies visual patterns from raw pixels. A CNN is often trained using methods like back propagation and gradient descent and has many layers of activations and convolutions dispersed among pooling layers (*Rawat & Wang, 2017*). Training a CNN model requires huge amount of data, computation time and processing power (*Miikkulainen et al., 2019*). Moreover, they are designed for solving a single specific task and have to rebuild from scratch once the feature space distribution changes. To overcome this isolated learning paradigm and utilizing the knowledge acquired from training for a single task, CNNs have advanced the idea of transfer learning: an optimization technique employing pre-trained models. A pre-trained model is a saved network that has previously undergone extensive training on a large dataset, typically for a large image classification task. They usually make use of ImageNet (*Deng et al., 2009*) and can be utilized directly in making predictions on new tasks or integrated into the process of training a new model. By using pre-trained models, both, the training time and generalization error are reduced.

Despite the remarkable success of deep learning, the lack of transparency and interpretability in their models has become a major concern among stakeholders. The deep learning models achieve a high level of accuracy, but being a black-box model it is almost impossible to identify the key features that led to the decision. In order to build trust and ensure accountability of the DL based systems, that are widely being deployed, there is a need for justification of decisions taken by models (*Samek, Wiegand & Müller, 2017*). Thus, there is a growing demand for understanding and interpreting the decisions undertaken by these black-box models. The need to verify the reliability of these networks is more emphasized recently after several cases were reported, where decisions taken by AI systems led to controversial consequences. A fatal accident by an Uber self-driving car (*Kohli & Chadha, 2019*), facial recognition systems evasion with a 3D printed mask (*Holmes, 2020*), gender biasing in the Amazon Recruitment tool (*Dastin, 2022*), and

bigotry in Correctional Offender Management Profiling for Alternative Sanctions (COMPAS) (*Wadsworth, Vera & Piech, 2018*) are some of the distressing automated decisions that accentuate the need for explanations of AI system.

Thus, the motivation of this study comes from the fact that deep learning models created from complex pre-trained models must be interpretable. Therefore, to evaluate the efficacy of these models in terms of interpretability, there is a need to apply XAI tools to verify the underlying reasoning of the model. Consequently, the hypothesis formulated for the research is "Complex deep learning models can be interpreted using Explainable AI techniques". In order to validate this hypothesis, we developed deep learning models for fruit classification task followed by the evaluation of interpretability of these models using the XAI technique. Fruit classification is a critical task in many industries such as agriculture, food processing and retail. Accurate fruit classification can help improve the efficiency of processes such as sorting, grading, and packaging, ultimately reducing waste and increasing profitability. Numerous studies have proposed deep learning-based model for fruit classification (*Hameed, Chai & Rassau, 2018*; *Khatun et al., 2020*; *Prakash & Prakasam, 2023*) but many of these models lack explainability, making it difficult to understand how the models arrived at their classification decisions. This can be a significant barrier to the adoption of such models in practical applications, particularly in industries such as agriculture or food processing. However, elucidating the inner workings of the models used in automated fruit classification systems will improve transparency, identify biases, and meet regulatory requirements that will eventually help users in trusting the model's output and increase their confidence in the system (*Rai, 2020*).

For the stated problem, we used a popular dataset Fruits 360 (*Oltean, 2021*) that comprises 90K images of fruits and vegetables with 131 different classes that makes it an ideal use case for our study. We utilized three pre-trained models VGG16, MobileNetV2, and ResNet50 as base network for three different classification models to classify specifically the type of fruit or vegetable. We fine-tuned and trained these models with Fruits 360 datasets to get an accuracy of around 98%. To ensure the transparency and reliability of our models we interpret these models by applying the widely used explainable AI tool, LIME.

The major contributions of this research are summarized below:

- We use transfer learning with three pre-trained models, VGG16, MobileNetV2, ResNet50 for classifying fruits and vegetables using complete Fruits 360 dataset comprising 131 classes.
- We employ two transfer learning techniques and evaluate the performance of pre-trained models.
- We compare the effectiveness of our models derived from pre-trained models using various evaluation metrics.
- We present a detailed explanation of our models using popular explainable AI tool LIME.

The remaining sections of the article are structured as follows: the Background and Related Work section presents the background and literature review. The Methodology

section explains the methodology adopted; the Experiments section presents the details of experiments performed. The Results and Discussion section discusses the results while the Conclusions section concludes the research with future directions.

## BACKGROUND AND RELATED WORK

Image classification has been around for decades. The conventional image classification models need large, labeled dataset, hand-crafted features, huge computation resources, and enormous time for training (*Desai, 2021*). This high cost of developing image classification models compromises their robustness. In contrast, deep learning models exploit multiple non-linear layers for feature extraction and classification, improve the efficiency of image classification task and have achieved astonishing results (*Vermeire et al., 2022*). Among several deep learning models, convolutional neural network (CNN) (*O'Shea & Nash, 2015*) has become the leading architecture for image classification and detection tasks such as facial recognition (*Khan et al., 2019*; *Mane & Shah, 2019*), medical image computing (*Gao, Lim & Jia, 2018*; *Zhang et al., 2019*), plant disease classification (*Lu, Tan & Jiang, 2021*), remote sensed image analysis (*Ma et al., 2019*).

Though CNN is the most popular deep learning model, its performance is largely dependent upon the volume of data. CNN requires a sizeable amount of data that needs plentiful computation power and time for training. Moreover, designing the CNN architecture from scratch is exhausting as it needs a lot of time and effort in finding suitable combination of layers and adjusted hyper-parameters. Further, the model learns for one specific task on a very specific dataset and has to rebuilt if the features-space changes (*Desai, 2021*; *Zhuang et al., 2020*). To overcome these challenges, researchers discovered the concept of transfer learning that allows transferring of knowledge gained by one task to solve similar another task (*Torrey & Shavlik, 2010*). It reduces the cost of training as the model is readily trained for identifying low level features and the last layer classifies the set of classes that were used during training. The following section discusses the use of transfer learning in various domains and the popular CNN architectures used in this study.

### Transfer learning

Deep learning approaches are typically dependent on the dataset used to train the network. A large volume of labeled data is needed to train a network to achieve desirable performance. Gathering a massive amount of labeled data for a particular domain is not only exhausting but also quite challenging in most real-world applications such as medical imaging (*Muhammad Dawud, Yurtkan & Oztoprak, 2019*). As a result, the idea of transfer learning has been introduced. Transfer learning allows utilizing the knowledge applied for solving one problem to resolve other relevant problems. The base network (commonly referred to as pre-trained network) is initially trained on a large dataset and transfers its learning parameters and weights to the target network. The last fully connected layer of the target network is then trained on its respective dataset. The pre-trained network can also be fine-tuned by retraining some of its layers to further increase performance. Transfer learning has been used widely in all machine learning applications, such as

computer vision (*Li et al., 2020*), natural language processing (*Ruder et al., 2019*), and speech recognition (*Qin, Qu & Zhang, 2018*), and it has demonstrated excellent results in terms of accuracy, training duration, and error rates.

In image classification, fine tuning a pre-trained model entails bootstrapping the top portion of the model, freezing the pre-trained convolutional layers and un-freezing the last few pre-trained layers. The frozen layers convolve visual features as usual while the un-frozen layers get trained on the custom dataset and updated according to the fully connected layer's predictions. ImageNet dataset is used for training for these pre-trained models as it encompasses around one million images belonging to 1,000 categories. Various CNN architectures have been developed as pre-trained models for image classification tasks; however, this research employs variants of three most popular architectures (VGGNet, MobileNet, and ResNet). The following sub-sections briefly introduces these CNN architectures.

### VGGNet

VGGNet (*Simonyan & Zisserman, 2014*) is a convolutional neural network with two variants *i.e.,* VGG16 and VGG19. VGG16 has 16 layers including 13 convolution and three fully connected layers, while VGG19 has 16 convolution layers and three fully connected layers supported by MaxPool layers. It is one of the prominent architectures in image classification. *Rezende et al. (2018)* used VGG16 to classify malware family by converting malware executable to a byteplot grayscale image and achieved 92.97% accuracy without any explicit feature engineering. *Kaur & Gandhi (2019)* applies similar methodology and classify MRI images as normal or abnormal with different neurological diseases by using VGG16 acquiring 100% accuracy. Comparison of different pre-trained models (VGG, ResNet, DenseNet, MobileNet, Inception, Xception) was presented by *Himabindu & Praveen Kumar (2020)*. They evaluated each model on accuracy, precision, F1-score, recall and reported that VGG outperforms all other models with 97% accuracy.

### MobileNets

MobileNets (*Sandler et al., 2018*) are convolutional neural networks designed by Google researchers. This CNN architecture is popular for its adoptability on mobile phones as it has a low resource requirement. The MobileNet architecture is developed using depthwise separable convolutions, which are lightweight deep neural networks that can have minimal latency for embedded and mobile devices. *Rabano et al. (2018)* uses MobileNet for classifying trash in an android application. *Shahi et al. (2022)* presented an attention-based MobileNetV2 architecture for fruit classification and evaluate it against accuracy, f1score, Kappa-score, WAFI, MAFI, recall and precision. The authors compared their architecture with other pre-trained models on three different datasets and their proposed framework surpassed all with achieving more than 95% accuracy on all three datasets.

### ResNet

ResNet (*He et al., 2016*) is a very deep residual network built by Microsoft and has a depth of 50 layers. ResNet combined multiple sized convolution filters which manage the

degradation problem and reduces the training time that occurs due to its deep structures. *Sarwinda et al. (2021)* proposed an image classification model for detection of colorectal cancer in colon glands images using different variants of ResNet and found that ResNet50 provides the most reliable performance for accuracy, sensitivity, and specificity. Precision classification for breast cancer histopathological image was investigated by *Jiang et al. (2019)* in which they proposed a customized version of ResNet. An accuracy of 99% was reported by the authors after the network was fined tuned.

The CNN architectures trained on large datasets have addressed the two major issues regarding the training of deep learning networks. These pre-trained models have reduced the requirement of voluminous data and the need for extensive computing environment to some extent. However, another critical concern with developing real-life deep learning systems is the need for an explainable, interpretable, and transparent solutions. Deep learning models remain black-box models and there is a growing demand for the explanation of their learning and prediction process. The following section discusses the new research paradigm known as Explainable AI (XAI) that is came into being to provide explanations of black-box models predictions.

## Explainable AI

Deep neural networks are regarded as black-box models by both developers and users since they are comparatively weaker in explaining their inference process and final decisions (*Xu et al., 2019*). Explainable AI is a collection of methods and techniques that allows end users to understand and trust the decisions AI systems make (*Holzinger et al., 2022*). It has gained significant attention recently among both industry and research community due to the fact that AI is now involved in such real-world applications that demands explainability and transparency, for example, medical diagnosis, investment recommendation, loan approvals, surveillance, autonomous vehicles, predictions for process optimization *etc.* Evaluation metrics such as high accuracy may not be sufficient to ensure that the decisions taken by these models are always correct, justified and without any bias. For example, COMPAS; an assistive tool used in multiple states of the US to assess the likelihood that a criminal offender would reoffend has been proven to be discriminatory, with results heavily biased towards white defendants (*Wadsworth, Vera & Piech, 2018*). Understanding the reasons behind the decisions taken by autonomous models leads to more reliable and trustworthy systems. Therefore, the goal of XAI is to make the reasoning behind the decision taken by AI systems that is understandable by humans (*Pearl, 2019*). A variety of XAI tools have been introduced to explain the predictions made by AI systems. Notably, LIME has emerged as a widely preferred choice.

## LIME

LIME (*Ribeiro, Singh & Guestrin, 2016*) is an acronym of Local Interpretable Model-Agnostic Explanations. It helps users understand why a machine learning model made a certain prediction by providing an explanation in terms of the most relevant features that influenced the prediction. To create an explanation, LIME generates a local, interpretable model around the instance being explained and weighs the contribution of each feature

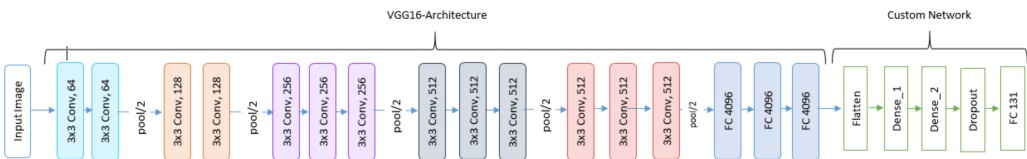

**Figure 1   DL architecture used in this study utilizing VGG16 pre-trained model.**

to the model's output. These features are then presented as an explanation to the user, in order to help them understand how the complex machine learning model arrived at the prediction. The explanations generated by LIME are intended to provide insights into the decision-making process of the black-box machine learning model, and to enable users to validate and understand the model's predictions. Many studies have been carried out that have used LIME for explaining the results irrespective of the nature of data (*Bhattacharjee et al., 2022*; *Zhu et al., 2022*; *Hamilton et al., 2022*). Consequently, we have used LIME for evaluating the interpretability of our fruit classification models in this research.

## METHODOLOGY

The methodology applied in this research encompasses two phases, that is; development of three classification models using transfer learning and explanation of these models using LIME.

In the first phase, three classification models are created using pre-trained models (VGG16, MobileNetV2 and ResNet50) as the base model. The features extracted from the base model are used by the new layers introduced in each model. However, the softmax layer is used as the last layer by initializing the number of neurons to total number of classes in Fruits 360 dataset *i.e.,* 131. Figs. 1, 2 and 3 depict the architecture of the classification models used in this study. The classification is performed by applying two transfer learning techniques, (a) using pre-trained models with frozen layers and (b) using pre-trained models with fine-tuned layers. Each classification model is evaluated through accuracy, precision, recall and F1-score.

In the second phase, the classification models are evaluated for their truthfulness. To accomplish this, five specific examples or instances are studied using LIME. The purpose of using LIME is to understand the underlying features that contribute to each decision made by the classification models. This helps in identifying what aspects of the data are driving the models' predictions thereby making it easier to understand the decisions taken by DL models.

## EXPERIMENTS

We analyzed the interpretability and understandability of three pre-trained deep learning models (VGG16, MobileNetV2 and ResNet50) by conducting experiments on fruit classification problem. We first developed fruit classification models using the pre-trained models and then employed explainable AI tool on classification models. For fruit classification problem, the Fruits 360 dataset has become a benchmark and has been

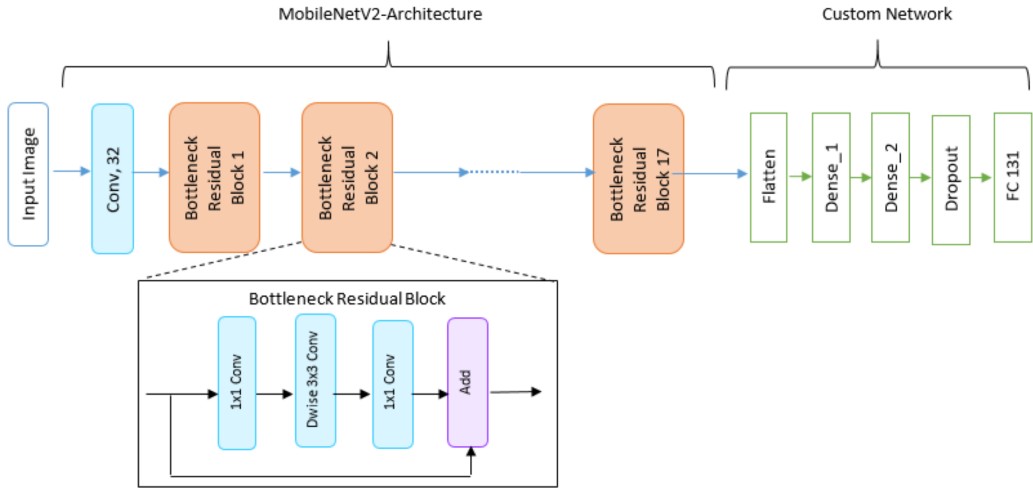

**Figure 2** DL architecture used in this study utilizing MobileNetV2 pre-trained model.

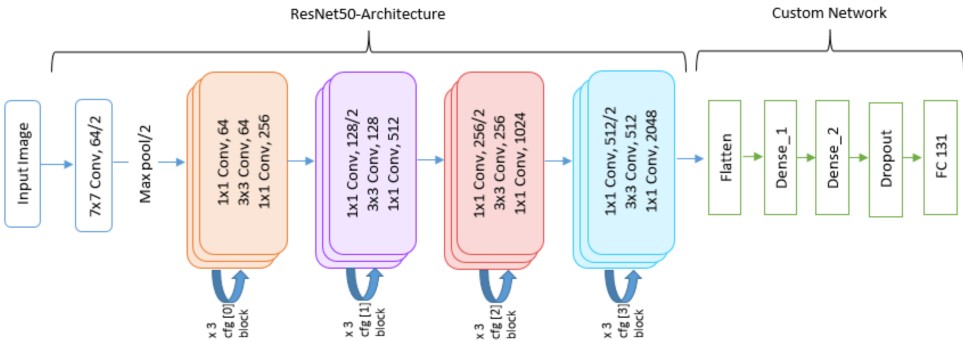

**Figure 3** DL architecture used in this study utilizing ResNet50 pre-trained model.

utilized in various research studies. *Ghosh et al. (2020)* used its 41 classes for detecting and classifying fruits using ShuffleNetV2. Similarly, *Siddiqi (2019)* has used 101 classes of Fruits 360 dataset and presented a comparative analysis on the performance of various deep learning models. Furthermore, *Sakib et al., (2019)* proposed a fruit recognition classifier by utilizing 17,823 images belonging to 25 categories. *Rathnayake et al. (2022)* have used the complete Fruits 360 dataset for image identification and recognition but similar to aforementioned studies they also have not taken the interpretability or explainability factor for their proposed models.

In this study, we have used a complete Fruits 360 dataset having 131 classes of fruits and vegetables images that is publicly available on Kaggle and developed classification models followed by their explanation. The dataset holds 90,483 images in total and each image

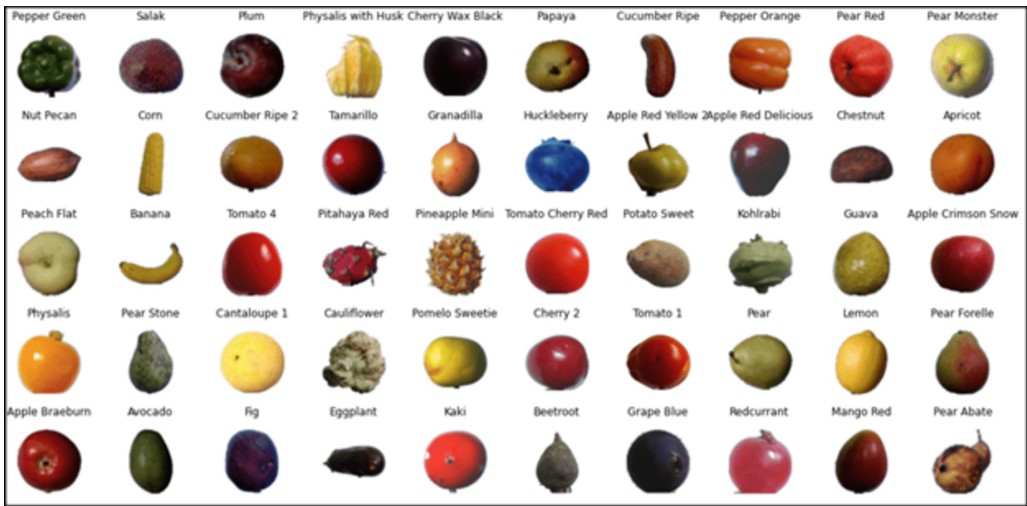

**Figure 4** Sample images from the Fruits 360 dataset.

has one fruit or vegetable. The size of each image is 100x100 and is captured with white background. Figure 4 shows some of the images from the Fruits 360 dataset.

The experiments were performed on the dataset using Keras API of tensorflow employing three pre-trained architectures, VGG16, MobileNetV2 and ResNet50. Two techniques of transfer learning, frozen and fine-tuned, were applied during the course of experiments to achieve the best possible results. In the frozen case, the pre-trained models convolve the data according to the ImageNet weights leaving their top portion, that is fully connected layer of the model, and transfer fixed features to the customized fully connected layer. While in fine tuning, the pre-trained models bootstrap the top portion, freeze the pre-trained convolutional layers and un-freeze the last few pre-trained layers. The frozen layers convolve visual features as usual while the un-frozen layers get trained on the custom dataset and updated according to the last fully connected layer. We used complete Fruits 360 dataset with 131 classes containing around 90K images; divided into training, validation, and test sets with a ratio of 65:10:25 respectively. Thus, the training set comprises 57,612 images; validation set contains 10,080 images while 22,688 images are part of the test set. *Naranjo-Torres et al. (2020)* and *Azarmdel et al. (2020)* have utilized the similar distribution of dataset for fruit image processing. All images in each set are pre-processed using the pre-processing function defined in the Keras Library for the respective pre-trained model. All experiments were carried out on a machine having an Intel Core i5-1135G7 @ 2.40 GHz processor with 32GB RAM and 2 GB NVIDIA GeForce MX350 GPU. The experiments performed in this study are presented in Table 1.

For frozen layer case, the training and validation batch size used is 64. The models are evaluated by training them iteratively with varying epochs that is 10 and 20 with training step size of 900 and validation step size of 157. The Adam optimizer is used from the Keras Optimizers and the initial learning rate was set to $10^{-3}$. Early stopping is applied in the validation loss with the patience value of 10 that is the model stops training if the validation

**Table 1  Experiments performed in the study.**

| Experiments | Results |
|---|---|
| **Phase I–Generation of Classification Models** | |
| Classification models development using pre-trained models with Frozen Layers | Table 3 |
| Classification models development by fine tuning the pre-trained models | Table 4 |
| **Phase II–Interpretation of Classification Models** | |
| Instances evaluated for LIME interpretations and their top 5 predictions | Table 5 |
| Top features selected by classification models for the chosen instances | Table 6 |

**Table 2  Hyperparameters used for development of Classification Models.**

| Hyperparameter | Value |
|---|---|
| Epochs | 10, 20 |
| Batch size (Training and validation) | 64 |
| Training step size | 900 |
| Validation step size | 157 |
| Learning rate | $10^{-2}$, $10^{-3}$ |
| Optimizer | Adam |
| Regularization | Early stopping |

loss keeps increasing till 10 successive epochs. For fine-tune case, the last two layers of each mode left un-freeze and the models are trained for 10 epochs with initial learning rate of $10^{-2}$. The hyperparameters used for the development of classification models are summarized in Table 2.

# RESULTS AND DISCUSSION

This section describes the results in terms of phases as mentioned in the methodology section. In the first phase, we generate classification models by utilizing three pre-trained models (VGG16, MobileNetV2, ResNet50) using two transfer learning techniques. While in the second phase, LIME interpretations are generated that highlight the key features used in the respective prediction. Table 3 reflects how our research objectives have been achieved with our corresponding research contribution.

## Phase I-development of classification models

The two techniques of transfer learning are frozen layers and fine-tuned layers as explained in the background and related work section. We have developed classification models by incorporating three pre-trained models VGG16, MobileNetV2 and ResNet50 whose results are described in successive sections.
**Table 3  Research objectives to research contribution mapping.**

| Research objectives | Contributions |
|---|---|
| To develop fruit classification models using pre-trained models | Three pre-trained models namely VGG16, ResNet50 and MobileNetV2 are utilized for the development of fruit classification models. See Figs. 5 and 6 |
| To explore transfer learning mechanism in the development of fruit classification models | Two transfer learning mechanisms (1) frozen layers and (2) fine-tuned layers are employed in the development of classification models. Refer to Section Phase I - Development of Classification Models for further details |
| To compare performance of fruit classification models using different metrics | The classification models employing pre-trained models are evaluated using accuracy, precision, recall and f1 score. See Tables 4 and 5 |
| To interpret fruit classification models using explainable AI tool | Explainable AI tool LIME is used to interpret the results of pre-trained models based classification models. See Table 7 and Fig. 8 |

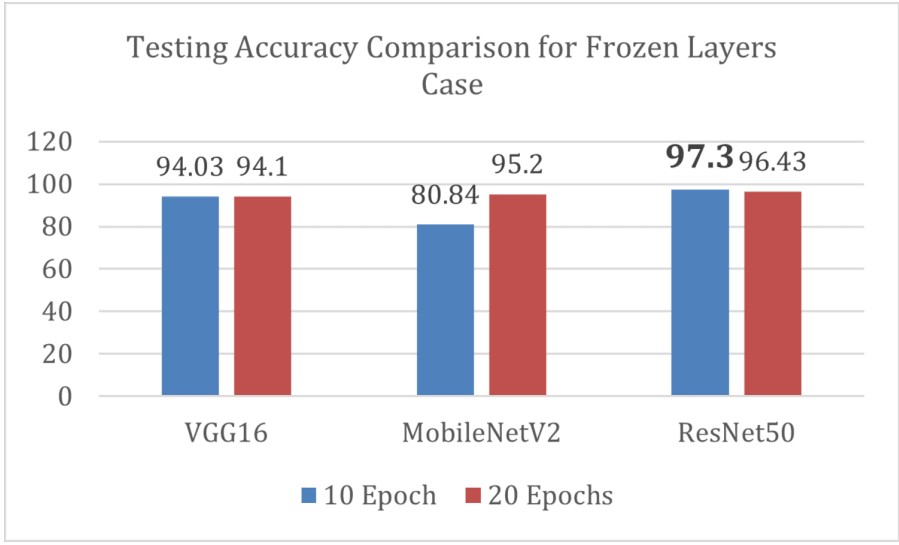

**Figure 5  Results of frozen layers technique of transfer learning.**

### Frozen layers

In frozen layers technique, where each pre-trained model served as a feature extractor with all layers freeze, ResNet50 outperformed VGG16 and MobileNetV2 models by achieving 97.3% test accuracy in 10 epochs as shown in Fig. 5.

The ResNet50 also outclasses the two models in terms of execution time as it takes comparatively less time to achieve highest accuracy. It is observed that VGG16 and ResNet50 start overfitting when set to train for 20 epochs while MobileNetV2 shows improvement by acquiring 100% training accuracy and 93.62% test accuracy. The results of experiments achieved by each model are presented in Table 4.

**Table 4  Classification models with frozen layers.**

| Parameters | VGG16 | | MobileNetV2 | | ResNet50 | |
|---|---|---|---|---|---|---|
| Epochs | 10 | 20 | 10 | 20 | 10 | 20 |
| Execution time (seconds) | 4,517 | 5,251 | 1,438 | 3,203 | 3,101 | 4,133 |
| Early stopped? | No | Yes, at Epoch 13 | No | No | No | Yes, at Epoch 14 |
| Training accuracy (%) | 99.8 | 99.9 | 98.2 | 99.99 | 99.8 | 99.9 |
| Validation accuracy (%) | 97.4 | 97.6 | 87.6 | 98.98 | 99 | 99.2 |
| Testing accuracy (%) | 94.03 | 94.1 | 80.84 | 95.20 | 97.3 | 96.43 |
| Precision | 0.95 | 0.96 | 0.83 | 0.95 | 0.97 | 0.97 |
| Recall | 0.94 | 0.94 | 0.8 | 0.95 | 0.97 | 0.96 |
| F1-Score | 0.94 | 0.94 | 0.8 | 0.95 | 0.97 | 0.96 |

**Table 5  Classification models with Fine-tuned Layers.**

| Parameters | 1-layer unfreeze | | | 2-layers unfreeze | | |
|---|---|---|---|---|---|---|
| | VGG16 | MobileNetV2 | ResNet50 | VGG16 | MobileNetV2 | ResNet50 |
| Execution time (Seconds) | 3,742 | 1,337 | 3,012 | 4,020 | 1,398 | 3,128 |
| Training accuracy (%) | 99.8 | 99.6 | 100 | 100 | 99.8 | 100 |
| Validation accuracy (%) | 97.9 | 96.6 | 99.9 | 99.1 | 96.1 | 99.8 |
| Testing accuracy (%) | 96.12 | 91.49 | 98.08 | 97.09 | 92.25 | 97.56 |
| Precision | 0.97 | 0.92 | 0.98 | 0.97 | 0.93 | 0.98 |
| Recall | 0.96 | 0.92 | 0.98 | 0.97 | 0.93 | 0.98 |
| F1-Score | 0.96 | 0.91 | 0.98 | 0.97 | 0.92 | 0.98 |

*Fine-tuned layers*

In fine-tuned technique, all models were trained for 10 epochs on two variations of unfreeze layers. First, all models are trained by tuning just one layer. Then, the models were trained by tuning their two layers in which all models got 100% training accuracy. The overall experiment results are presented in Table 5.

VGG16 and MobileNetV2 show relatively improved performance by achieving 97.09% and 92.25% test accuracy, respectively after unfreezing their last two layers. However, ResNet50 dominance is witnessed again on VGG16 and MobileNetV2 as it attains test accuracy of 98.08% by unfreezing one last layer only as depicted in Fig. 6.

It is evident from analyzing the experimental results that pre-trained models offer greater accuracy in comparatively smaller training time. ResNet50 exhibits highest accuracy in both cases while VGG16 and MobileNetV2 have also shown greater than 90% accuracy which is acceptable in classification problems. The highest accuracy produced version of each pre-trained model is used for the investigation of their interpretation capability using LIME.

## Phase II-Interpretation using LIME

LIME produces instance level explanations, therefore five correctly predicted instances were selected for interpretations' study. Table 6 presents the instances and their top five predictions made by each model.

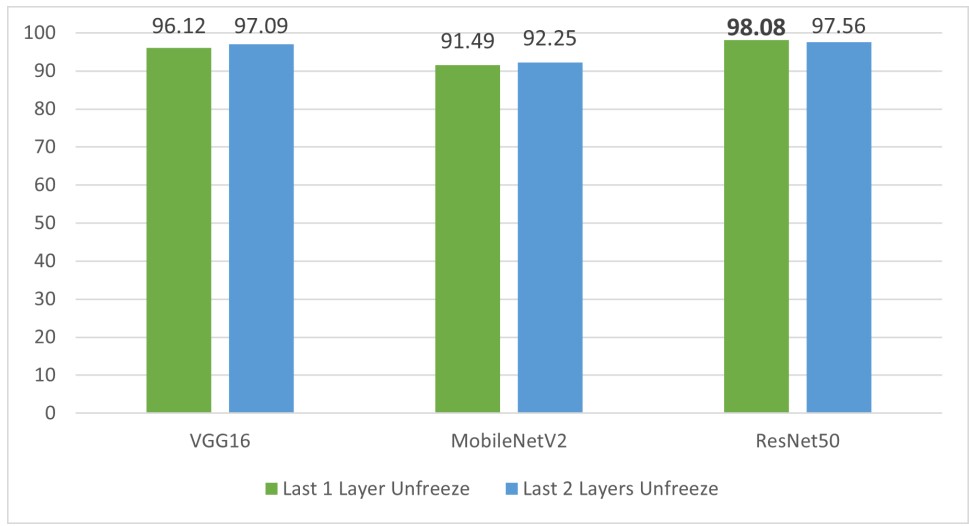

**Figure 6** Results of fine-tuned layers technique of transfer learning.

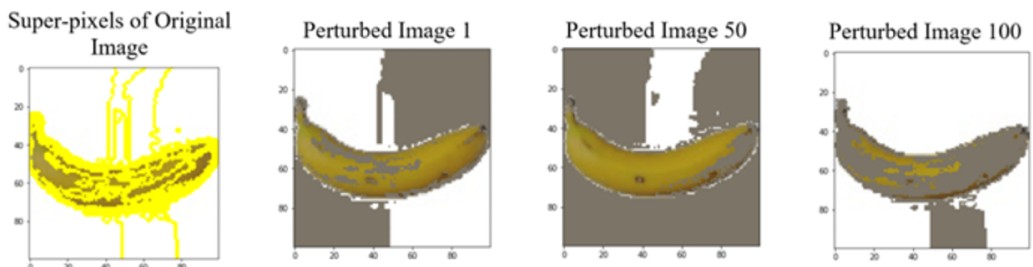

**Figure 7** Perturbed images.

LIME works by creating perturbed images of the instance being predicted; therefore, 150 perturbed images were generated by turning on and off the super pixels of the instance as depicted in Fig. 7. Super pixels are used in LIME to help generate explanations for image classification models by simplifying the image and reducing the number of features used to explain the model's prediction. Instead of considering every individual pixel in an image, LIME groups pixels into super pixels and treats each super pixel as a feature. The left of the figure shows the image with all super pixels on while the right of the image shows some of the perturbations generated by the ResNet50 model. Similar perturbed images were generated by MobileNetV2 and VGG16.

Each perturbed image is then predicted and the distance between the perturbed image and the original image is calculated. Cosine metric was used to find the distance with kernel size of 0.25. Finally, the linear regression model is fitted to find out the top feature in the model prediction. Figure 8 shows the top feature selected in green color by the models to make prediction of the instance chosen by each model. It is clearly evidenced that VGG16

**Table 6  Instances evaluated for LIME interpretations.**

| Instance chosen | Top 5 predictions | | |
|---|---|---|---|
| | **VGG16** | **MobileNetV2** | **ResNet50** |
|  | Banana<br>Peach 2<br>Physalis<br>Banana Red<br>Tomato Maroon | Banana<br>Kaki<br>Banana Red<br>Carambula<br>Banana Lady Finger | Banana<br>Banana Lady Finger<br>Carambula<br>Lemon<br>Cucumber Ripe 2 |
|  | Pineapple Mini<br>Tomato Yellow<br>Tomato Maroon<br>Apple Red Yellow 1<br>Huckleberry | Pineapple Mini<br>Physalis with Husk<br>Mulberry<br>Rambutan<br>Pitahaya Red | Pineapple Mini<br>Pineapple<br>Mangostan<br>Mulberry<br>Rambutan |
|  | Apple Braeburn<br>Apple Red 2<br>Apple Red 3<br>Tomato Yellow<br>Apricot | Apple Braeburn<br>Nectarine<br>Apple Red 2<br>Apple Pink Lady<br>Tamarillo | Apple Braeburn<br>Apricot<br>Potato Red Washed<br>Nut Forest<br>Cherry Wax Yellow |
|  | Pepper Green<br>Tomato not Ripened<br>Watermelon<br>Eggplant<br>Grape Pink | Pepper Green<br>Apple Red Yellow 2<br>Tomato not Ripened<br>Tomato Heart<br>Pepper Orange | Pepper Green<br>Pepper Red<br>Eggplant<br>Dates<br>Tomato Heart |
|  | Strawberry<br>Cucumber Ripe<br>Grape Pink<br>Raspberry<br>Dates | Strawberry<br>Mandarine<br>Nectarine<br>Lemon<br>Strawberry Wedge | Strawberry<br>Cucumber Ripe<br>Avocado ripe<br>Strawberry Wedge<br>Carambula |

and ResNet50 select the most suitable position of the image to be predicted as banana. Whereas MobileNetV2's feature selection is abstruse.

Table 7 shows the top feature selected for the rest of the instances presented in Table 6 by each model. Nevertheless, it is apparent that the predictions of the pre-trained model are explainable, interpretable and can be trusted as the top feature chosen by each model is suitable for the respective data instance. Hence, our hypothesis that complex deep learning models can be interpreted using explainable AI techniques has been successfully validated by these findings.

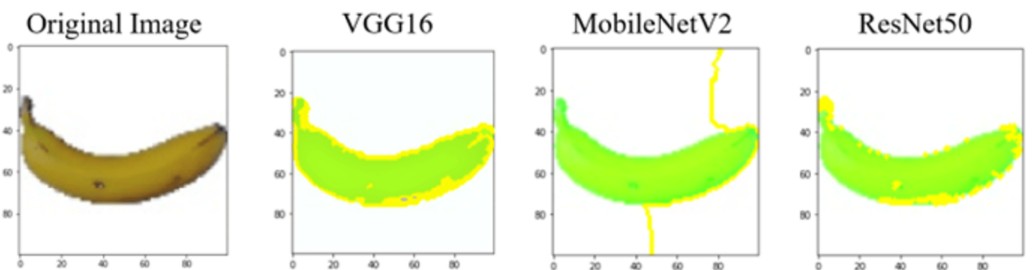

**Figure 8  Top feature selected by each classification model for the chosen instance.**

**Table 7  Top feature selection for the prediction for each instance.**

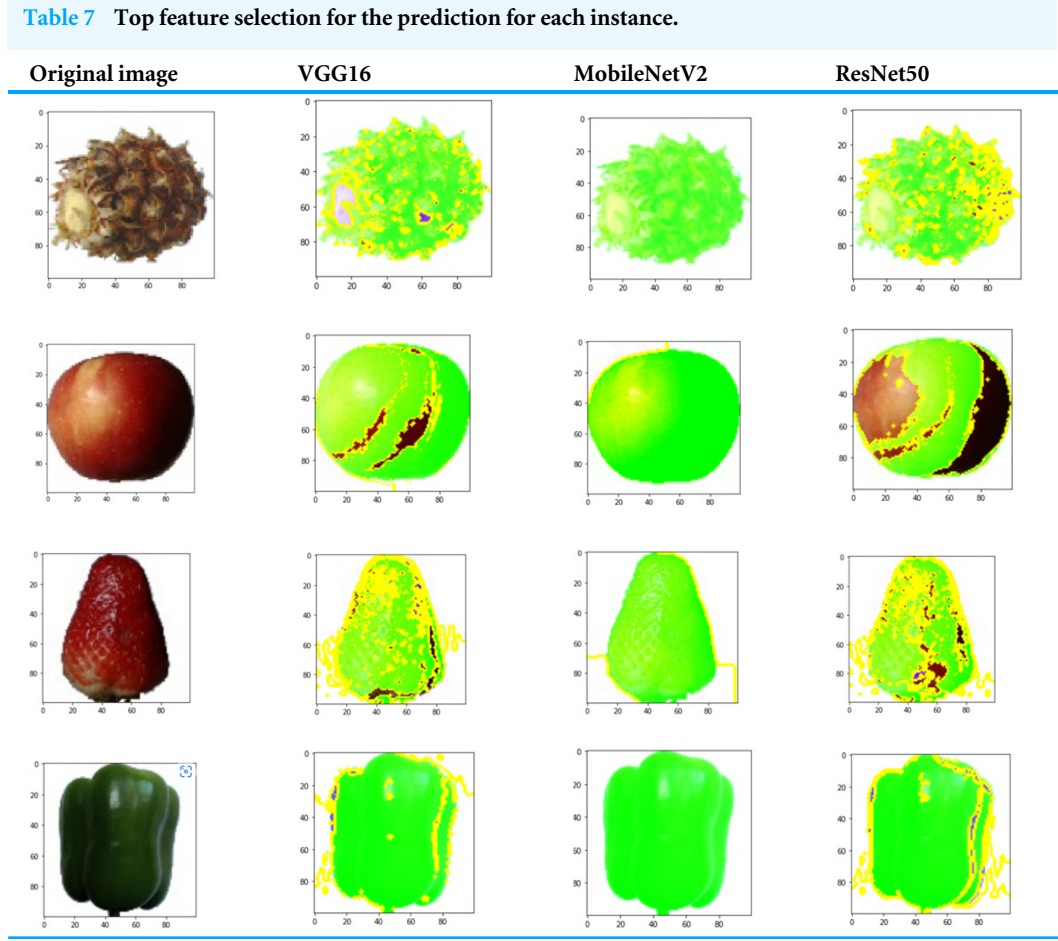

| Original image | VGG16 | MobileNetV2 | ResNet50 |
|---|---|---|---|

## CONCLUSIONS

In this study, our objective was to investigate the interpretability of deep learning models due to their black-box nature that leads to a lack of transparency and interpretability in their decision-making process. Although these models have shown exceptional accuracy, the absence of justification raises concerns related to trust, accountability, biases, and transparency. Therefore, we aim to address this issue by exploring the interpretability of

deep learning models using explainable AI techniques. We assessed the truthfulness of the deep learning models specifically pre-trained models by generating interpretations of their predictions. To do this, we performed experiments in two settings. First, we utilized three pre-trained models (VGG16, MobileNetV2 and ResNet50) with two techniques of transfer learning. Second, we produced interpretations of these models using LIME. Extensive experiments have been carried out to obtain the classification model with highest accuracy. ResNet50 has outperformed VGG16 and MobileNetV2 by attaining 98.08% accuracy with one last layer unfreeze. While VGG16 and MobileNetV2 have shown significant performances with 97.09% and 93.62% accuracy, respectively. Few instances that are correctly predicted by the models are selected to evaluate the interpretability of each pre-trained model. LIME explanations are generated that mark the top feature selection in the underlying prediction and it is observed that all pre-trained models used in this research are interpretable.

As future perspective, we intent to extend our research on the interpretability of deep learning models into more critical and sensitive applications such as healthcare and finance. The interpretation of these models is essential to ensure transparency and enhance trust in these domains. Additionally, we aim to investigate the root cause of any misclassifications made by the models using various explainable AI techniques. This will allow us to identify the areas that need improvement and develop strategies to reduce misclassifications in classification models. By doing so, we can improve the performance and accuracy of deep learning models, making them more reliable and trustworthy in critical applications.

### Funding
The authors received no funding for this work.

### Competing Interests
The authors declare there are no competing interests.

### Author Contributions
- Tehreem Qamar conceived and designed the experiments, performed the experiments, analyzed the data, performed the computation work, prepared figures and/or tables, authored or reviewed drafts of the article, and approved the final draft.
- Narmeen Zakaria Bawany conceived and designed the experiments, analyzed the data, authored or reviewed drafts of the article, and approved the final draft.

### Data Availability
The data is available at Kaggle: Available at https://www.kaggle.com/datasets/moltean/fruits.

The code is available at GitHub and Zenodo: Available at https://github.com/tehreemq/FruitClassification.git

tehreemq. (2023). tehreemq/FruitClassification: First Release (v1.0). Zenodo. Available at https://doi.org/10.5281/zenodo.7953449.

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
