# Peer review of "Understanding the black-box: towards interpretable and reliable deep learning models"

_PeerJ Computer Science, doi:10.7717/peerj-cs.1629_

## Round 0.1 · original submission · Major Revisions

Please revise your paper according to the reviewers' comments.
Thanks.

Reviewer 1 ·

Basic reporting

See the general suggestions section

Experimental design

See the general suggestions section

Validity of the findings

See the general suggestions section

Additional comments

General suggestions:
Suggestions for this part "Introduction":
The introduction should be rearranged and unnecessary literature removed.
Nowhere did I see AI (DP) related to the title of this paper.
Search for literature on your topic.
You should therefore reorganize your Introduction so that it is clear:
- what is the topic you cover in the paper
- what we already know about that topic,
- what we do not know (what the knowledge gap is)
- why it is important to fill this gap
- how do you want to fill this gap
- what are your results and how do they contribute to advancing our knowledge about this topic?
Missing citation

Suggestions for this part "Background and Related Work"
You should present existing research on how fruit classification through explainable AI is fruitful, as this is the core topic of your paper.
In my opinion, you should not simply present a list of statements from various authors. Instead, you should try to illustrate your problem (what affects the fruitfulness of fruit classification using explainable AI) and systematize the contributions provided by other researchers to this question.
I think that in this section you should organize existing research on the topic of your paper and try to present a clear picture of the results of previous research about factors influencing your topic. Is Fruit Classification really Fruitful? Evaluating Fruit Classification Models using Explainable AI

The construction and elaboration of hypotheses are missing. So, also in this regard, I see a lack of consistency between the objectives you state you will pursue and the analysis you present. I also have some concerns about how you present your empirical analysis:
- you present not present hypotheses, and you do not explain how you arrived at them and what is the connection with our existing knowledge on the subject. Hypotheses must be formulated with a deductive approach, i.e. result from the knowledge we already have on this topic;
- the connection between your hypotheses and your research objectives (the ones you have indicated in the previous pages) is not very clear;
Again, the suggestions are the same as in the introduction that I highlighted. The link to your paper title is missing. Please rearrange it and base it on the variables taken in the study.
Suggestions for this part "Experiments"
Who has made a similar contribution regarding this part that you have highlighted?

Suggestions for this part "Results and Discussion"
You should also explain why you think that the method you propose is aligned with your research goals. On the contrary,
Implications for theory and practice are very weak and insubstantial. In your discussion of the results, you should make it clear how they enrich our knowledge of the impact of your topic. I did not see the findings of this research in other research. Comparability is lacking
Suggestions for this part "Conclusions"
deep rearrangement

English language edition

Reviewer 2 ·

Basic reporting

The paper is written satisfactorily, but there are some portions that need to be further improved:

- Various strengths of deep learning (DL) models are given, for example: `Unlike traditional machine learning models', where they `have the capability to automatically engineer features ... produce ample accuracy'. These claims need to be backed up with the appropriate reference(s).

- A key feature of the methodology, LIME, is mentioned before the contributions on line 81. The term is however not described until much later; some exposition needs to be given as to what LIME is (much like VGG16, etc are already introduced as pre-trained models).

- In the subsection on Explainable AI (XAI), LIME is described as being able to `provide explanations' - what are these explanations? This is not properly quantified or described throughout the paper.

- Furthermore, two other XAI methods (Grad-CAM and SHAP) are described in detail, but not used. These are superfluous to the paper and should be removed. The focus should be on LIME

- Figures 1--3 are very blur and difficult to read.

Experimental design

- In line 78, the Fruit360 dataset is said to contain 10K (sic - kilo is not capitalised) images, while in line 229, the Fruits 360 dataset is said to have 90 483 images. Which spelling and figure are the correct ones?

- It is not clear how the parameters of the study (e.g., the size of the training, validation, and test sets, the hyper-parameters) were chosen. This is important to justify the experimental method so effects on the results can be explained.

- What does `super pixel' in line 306 mean?

Validity of the findings

The paper is the simple application of existing techniques (pre-trained datasets, established XAI tools) onto a well-studied problem (the classification of fruits). It is difficult to quantify how well the method performs, as no controls were used and no comparisons with other existing methods are made.

Reviewer 3 ·

Basic reporting

This article tests the ability of the AI to interpret the images it encounters. The authors succeded in reviewing what is already known in this major, and transitioning to the gap in knowledge they inteded to fill in the present work.

Experimental design

the authors thoroughly described the methods they used, which makes it easy to the readers who are not familiar with the concepts that are discussed, to understand the work that is presented.
Some of the methods were mentioned in the introduction. Ethical consideration was not mentioned, neither was the funding of the study. The software that the authors used to analyse the results was not mentioned.

Validity of the findings

the authors supllied the article with generous figures and tables, but it would be easier to insert the figures in the bulk of the article to facilitate the reading flow. The figures have low resolution..

---

## Round 0.2 · Major Revisions

The Academic Editor who originally handled this submission is no lopnger available and so I have taken over. Please carefully check the comments from the reviewer to make sure all the comments have been thoroughly addressed in the revised manuscript accordingly.

Reviewer 2 ·

Basic reporting

Many of my comments have been addressed well, but some issues still remain:

- The flow of the motivation is now jarring as it goes from cases requiring explainable AI (XAI) to LIME and suddenly to using LIME for the fruit classification problem. LIME as a XAI tool should be introduced separately from the fruit classification problem.

- The metric suffix kilo is not capitalised (i.e., 90k).

- The portion on XAI in the Background and Related Work section is still worded to introduce several methods, as opposed to only LIME in this work.

Experimental design

no comment

Validity of the findings

While the paper presents a set of results using LIME as XAI for the fruit classification problem, it is unclear how well LIME itself performs as XAI if there are no comparisons with other existing methods applied to the same problem. Thus while it can be stated that LIME was successfully applied, it cannot be said that it performed well.

---

## Round 0.3 · accepted · Accept

The remaining comments have been addressed satisfactorily.

Reviewer 2 ·

Basic reporting

My comments have been sufficiently addressed. As a note, it should be `90k' (not `90K' as is currently written).

Experimental design

no comment

Validity of the findings

no comment